# Global Proteome Profiling Revealed the Adaptive Reprogramming of Barley Flag Leaf to Drought and Elevated Temperature

**DOI:** 10.3390/cells12131685

**Published:** 2023-06-22

**Authors:** Krzysztof Mikołajczak, Anetta Kuczyńska, Paweł Krajewski, Michał Kempa, Natalia Witaszak

**Affiliations:** Institute of Plant Genetics, Polish Academy of Sciences, 60-479 Poznań, Poland; akuc@igr.poznan.pl (A.K.); pkra@igr.poznan.pl (P.K.); mkem@igr.poznan.pl (M.K.); witaszak.natalia@gmail.com (N.W.)

**Keywords:** abiotic stress, combined stress, flag leaf, proteomics, regulatory components, stress-induced proteins

## Abstract

Plants, as sessile organisms, have developed sophisticated mechanisms to survive in changing environments. Recent advances in omics approaches have facilitated the exploration of plant genomes; however, the molecular mechanisms underlying the responses of barley and other cereals to multiple abiotic stresses remain largely unclear. Exposure to stress stimuli affects many proteins with regulatory and protective functions. In the present study, we employed liquid chromatography coupled with high-resolution mass spectrometry to identify stress-responsive proteins on the genome-wide scale of barley flag leaves exposed to drought, heat, or both. Profound alterations in the proteome of genotypes with different flag leaf sizes were found. The role of stress-inducible proteins was discussed and candidates underlying the universal stress response were proposed, including dehydrins. Moreover, the putative functions of several unknown proteins that can mediate responses to stress stimuli were explored using Pfam annotation, including calmodulin-like proteins. Finally, the confrontation of protein and mRNA abundances was performed. A correlation network between transcripts and proteins performance revealed several components of the stress-adaptive pathways in barley flag leaf. Taking the findings together, promising candidates for improving the tolerance of barley and other cereals to multivariate stresses were uncovered. The presented proteomic landscape and its relationship to transcriptomic remodeling provide novel insights for understanding the molecular responses of plants to environmental cues.

## 1. Introduction

It has been projected that climate change will intensify the occurrence of environmental stresses, including drought and heat [1]. This will, in turn, lead to a serious reduction in plant productivity. Counteracting this remains a great civilizational challenge and a particular focus of scientists [2]. There is a strong need for effective tools for improving plant performance under the increasingly severe and frequent adverse conditions expected in the near future.

The response to environmental stress is manifested at the whole-plant level. Genome- and transcriptome-wide studies performed across plant species have undoubtedly been groundbreaking in terms of boosting our understanding of the molecular basis of plant functioning, including reactions to environmental hazards, and for identifying candidate genes [3]. However, it should be borne in mind that such information alone may be insufficient, since translational and post-translational processes play key roles in ensuring organism survival [4]. Thus, the enrichment of transcriptomics by illustrating proteome remodeling appears to be a milestone in deciphering the mechanisms and pathways of plant responses to stress. Recent advances in omics approaches and an increase in their throughput have resulted in numerous studies on the molecular response of plants to environmental cues, including barley [5]. Overall, proteome-wide quantification and interpretation are now more feasible, such as for global analysis of gene expression at the protein level.

Over the years, the action of numerous stress-induced proteins has been reported in various plant species. This includes enzymes (e.g., proteins participating in detoxification), transcription factors (TFs), transport and signaling proteins, and membrane receptors [6,7,8]. Kinases and phosphatases also play crucial roles in plant responses to an adverse environment through phosphorylation/dephosphorylation events [9]. Late embryogenesis-related proteins (LEAs) are among the most widely examined protective agents against drought and other stressors [10]. In addition, heat shock proteins (HSPs)—molecular chaperones—have been well documented to preserve the stability of other proteins upon stress and to assist in maintaining membrane integrity by interacting with lipids [11,12]. Interesting research was performed by Rodziewicz et al. [13], who used a mapping population of barley recombinant inbred lines exposed to drought to integrate genotypic and proteomic data. They detected several quantitative trait loci linked to proteins (pQTL) involved in defense mechanisms, for instance, HSP70 (on chromosome 4H) and betaine aldehyde dehydrogenase (on chromosome 1H).

Notably, most of the reports have focused on the influence of a single stress on plants; however, being sessile organisms, plants are usually exposed to several abiotic stresses rather than just one; for example, drought is often accompanied by an elevated temperature [14]. Hence, the extensive investigation of molecular responses to combined stresses is of great importance. Research conducted on barley is of particular value since it is now being used as a model system for genetic/molecular studies in cereals [15,16]. Moreover, barley is widely known to be suitable for cultivation in marginal habitats [17]. In recent years, significant progress has been made in unraveling the genome of barley, which has accelerated our understanding of the molecular background of its behavior in a changing environment [18,19]. Unfortunately, there is only limited literature on molecular characterization of the barley’s flag leaf, whose vitality under stress is fundamental for grain yield formation. Its effective photosynthesis enables plentiful assimilates to be transported into the spike, especially during the grain filling period; decreased grain yield due to a destroyed flag leaf has been confirmed [20,21]. Recently, Mikołajczak et al. [22] presented pioneering profiling of barley flag leaf transcriptomes under drought and heat stress, revealing potential targets for improving the ability of cereals to withstand multiple abiotic stresses.

The objective of this study was to obtain a better understanding of plant responses to combined abiotic stresses through unraveling the genome-wide changes in protein accumulation in barley flag leaf exposed to drought, elevated temperature, or both. Proteins whose expression was induced universally or by a specific stress treatment were identified. It was also assessed whether the proteomic modifications depended on the particular flag leaf morphology. Moreover, we confronted the results of gene expression at the proteomic and transcriptomic levels and constructed a network of substantially correlated response profiles of differentially expressed genes and proteins.

## 2. Materials and Methods

### 2.1. Experimentation

The plant material, experimental conditions, and sample collection were as described in detail in our previous report [22] on a study in which phenotypic, physiological, and transcriptomic analyses were carried out. This previous experiment also provided the material for large-scale proteomic examination in the present study. Briefly, seven spring barley recombinant inbred lines (hybrids of European and Syrian accessions; see [23,24]) were classified into three groups according to the size of the flag leaf (small—S, medium—M, large—L) (Appendix A). Five flag leaves of each genotype (seven genotypes in total) were used to define the average dimensions and calculate the leaf area (of the rectangle framing the leaf) as described in our previous study by Mikołajczak et al. [22]. Overall, four environmental variants were applied, namely, control conditions (C) and three stress treatments: heat (H), drought (D), and their combination (HD). The optimal/control conditions were as follows: air temperature at 8/14 °C and 12/12 h (night/day) photoperiod for 1 month, followed by 16/22 °C and 8/16 h (night/day) photoperiod until the end of the vegetation season; the soil moisture was kept above 70% of field water capacity (FWC). Abiotic stress conditions were as follows: 20/30 °C (night/day) and FWC as in C for H treatment, 20% FWC and temperature as in C for D treatment, and 20/30 °C (night/day) and 20% FWC for HD. Before and after the application of stress, plants were grown in control conditions. Flag leaves for protein extraction and quantification were sampled at two time points, namely, the third (T1) and seventh (T2) days of stress. Three biological replications were used in the study and each replication comprised at least three flag leaves collected from plants in one pot; in total, 168 samples were collected for proteomic analysis.

### 2.2. Protein Extraction and Quantification

Sampled leaves were immediately frozen in liquid nitrogen and stored at 80 °C until analysis. Samples were homogenized in liquid nitrogen and then 100 mg of powder (per replication) was used for protein extraction, in accordance with the protocol of Hurkman-Tanaka [25]. Precipitated protein was dissolved in 50 mM ammonium bicarbonate and quantified using PierceTM BCA protein Assay Kit (Thermo Fisher Scientific, Waltham, MA, USA). Protein samples, at a concentration of 1.0 mg/mL, were subjected to “in-solution” digestion with trypsin solution (Sequencing Grade Modified Trypsin, Promega, Madison, WI, USA) overnight. Just before digestion, peptide solution was pre-treated with 100 mM DTT for 5 min at 95 °C and 100 mM iodoacetamide for 20 min at an ambient temperature.

Untargeted proteomic analyses were performed using a Dionex UltiMate 3000 RSLC liquid chromatograph coupled to a Q Exactive high-resolution mass spectrometer with an Orbitrap mass analyzer equipped with an H-ESI ion source (Thermo Fisher Scientific). Two mobile phases were used, consisting of (A) water containing 0.1% formic acid (*v*/*v*) (LC-MS grade, Merck, Darmstadt, Germany) and (B) acetonitrile (LC-MS grade, Merck), with the application of the following linear gradient: 0 min—5% B, 5 min—5% B, 160 min—70% B, 160 min—95% B, 170 min—95%, 170 min—5% B, and 180 min—5% B. For the chromatographic separation, 5 µL of sample was injected onto an Acclaim PepMap RSL C18 column (75 µm × 250 mm, 3 µm, Thermo Fisher Scientific) and its temperature was maintained at 30 °C. The applied flow rate was 0.3 µL/min. A mass spectrometer was operated at a positive-ion mode spray voltage of 1.5 kV, capillary temperature of 250 °C, and S-lens RF level of 50.0. Full MS followed by ddMS2 mode was employed. The full scan covered the *m*/*z* range of 350–2000 and was performed at the following settings: −70,000, AGC target–5 × 10^6^ and maximum IT—100 ms. The settings of the data-dependent mode were as follows: resolution—17.500, AGC target—5 × 10^4^, maximum injection time—100 ms, isolation window—2.0 *m*/*z*, and normalized collision energy. The system was operated using Xcalibur 4.0 software (Thermo Fisher Scientific). Out of 168 samples collected for protein quantification, 154 were successfully analyzed (Appendix A).

Data were processed using MaxQuant (1.5.3.1) and Perseus (1.4.1.3) software, as well as the commercial software Proteome Discoverer 2.2 (Thermo Fisher Scientific). Protein libraries were searched using the SequestHT tool for proteins available for *Hordeum vulgare* in the UniProt database. Out of 2028 proteins found, data on 1431, in the form of scaled abundances (Appendix A), were subjected to statistical analysis after data quality control based on comparison of the abundances in control extracts.

### 2.3. Data Analysis

Protein abundances were transformed logarithmically to base 2 before being subjected, for each protein independently, to analysis of variance in a mixed linear model containing the following fixed effects: experimental variants (i.e., combinations of leaf size groups and time points, 6 levels), stress treatments (4 levels), and their interaction, along with random effects of genotypes within variants. Proteins for which more than 25% of data were missing were not analyzed. Comparisons of treatments H, D, and HD vs. the control were performed for all six variants (18 “contrasts”). A protein was defined as differentially expressed if the absolute value of the contrast estimate, which was equivalent to the log2 (fold change) value, was greater than 1.5, and the mean treatment effect or the (variant × treatment) interaction effect was significant for that protein in ANOVA (*p* values corrected by the Benjamini–Hochberg method < 0.05). Principal coordinate analysis was based on the matrix of Euclidean distances. All of the above analyses were performed in Genstat 19 [26]. A weighted co-reaction network analysis was performed using the library WGCNA in R [27,28]. For proteins, the network parameters were as follows: beta = 4, complete link clustering method, cutHeight = 0.80, and minsize = 5; for proteins and genes, the parameters were: beta = 6, complete link clustering method, cutHeight = 0.98, and minsize = 10. Venn diagrams were made using library venn 1.11 in R. The assignment of barley genes from the IBSC_v2 Hordeum vulgare (Ensembl Plants rel. 48) genome assembly to proteins was performed by the UniProt TrEMBL identifier present in the gene annotation. Identifiers of genes corresponding to IBSC_v2 sequences in the MorexV3 pseudomolecule assembly (Ensembl Plants rel. 52) were found using blast of coding sequences of genes (e-value < 1 × 10^−50^).

## 3. Results

### 3.1. Differentially Expressed Proteins

Out of 1431 identified proteins, 413 were expressed differentially in at least one of the 18 defined comparisons (Appendix A), and in all cases the number of upregulated proteins was larger than that of downregulated ones (excluding group M in D, T2) (Table 1).

The greatest number of differentially expressed proteins (DEPs) was observed for plants of group L, whereas it was the lowest for group M, (approximately half as many DEPs as in group L). Consequently, genotypes of groups L and S had more DEPs in common (approximately 1.5-fold) than in any other comparison of groups, and 58 DEPs were shared among all defined groups with different flag leaf sizes (Figure 1A). Overall, drought as well as its combination with heat induced more changes (almost 1.5-fold) in protein accumulation than heat alone (Figure 1B). However, in some cases, especially for group L, H affected the regulation of more or an equal number of proteins compared with HD at T1 and with D at T2, respectively (Table 1). The numbers of DEPs shared between D and H as well as between H and HD were similar, but these numbers were lower than that of DEPs in common between D and HD; 115 DEPs were shared across all stress variants. More than twice as many proteins changed their regulation status exclusively in response to D and to HD than to H (Figure 1B). In general, changes in protein expression were less numerous at the early time point than at the late one, but notably, approximately 60% of DEPs at T1 were also identified at T2 (Figure 1C); exceptionally, in group L, the number of DEPs in D at T1 was higher (and equal for H) than at T2, similar to the findings in group S in H for upregulated proteins (Table 1).

The identified DEPs were annotated using Gene Ontology (GO) terms. To identify for which GO terms the assigned DEPs were especially up- or downregulated in response to stress stimuli, we calculated the marginal distribution (356 downregulation/794 upregulation) of all proteins showing differential expression events (DE events) in the experiment and identified DEPs with the largest deviations from the marginal distribution (Figure 2). DE events assigned to the term “response to stimulus” corresponded mostly to protein upregulation in response to applied stresses, similar to those assigned to the term “cell organization and biogenesis”. In turn, the numbers of up- and downregulated proteins within the remaining GO terms were rather balanced (more than in the marginal distribution), such as for “transport”.

Interestingly, out of the DEPs annotated to “response to stimulus”, three (A3RHE3, A0A0S1LGH4, Q5D5Z6) were found to be dehydrins (Appendix A). In general, they were upregulated in D and HD at T2 across the genotypes of different flag leaf sizes; however, the protein A3RHE3 was downregulated in genotypes with a small flag leaf exposed to HD at T2 but upregulated in D at T1. In addition, in group L, the regulation of this protein was modified only at T1 (in H and HD). Increased accumulation of dehydrin Q5D5Z6 was observed in HD at T2 for all groups of genotypes (and in D, T1 in group S), and additionally, this DEP was found across all stress treatments in group M at T2 (and in HD at T1). A similar profile of changes was identified for dehydrin F2DH00 with the difference that the DEP was in common among all stress variants in group L at T2, instead of group M. Three DEPs assigned to the term “response to stimulus” were also annotated to “antioxidant activity” in the molecular function category. They showed various regulations in defined comparisons (between groups of different flag leaf sizes, treatments and time points), but it can be pointed out that protein F2DBE3 (catalase) changed its expression status from negative at T1 to positive at T2 in response to D in genotypes of group S. Among the proteins that function in transport, we found that ferredoxin (Fd) (M0VYW3) and chloride channel 1 (E9LFE6) were up- and downregulated, respectively, in both D and HD at T2 in genotypes of group S. In turn, aquaporin HvPIP1;3 (O48518), cytochrome subunit beta (A0A218LNQ7), and predicted protein F2CUI8 (mitochondrial carrier protein according to Interpro) had increased accumulation in group L across the stress treatments at T1; meanwhile, all of them (excluding O48518) were downregulated in group M in D at T2. Three proteins annotated to photosystem II, namely, D2 protein (A0A218LNI3), cytochrome b6 (P60161), and cytochrome b559 subunit alpha (A0A218LNB6), responded specifically (downregulation) to HD in group L at T2 (Appendix A). Additionally, four DEPs related to photosystem I or II reaction center were detected. Proteins A1E9V0 and the above-mentioned A0A218LNI3 (both in photosystem II) were exclusively downregulated in group L in HD at T2, whereas P13194 and Q00327 (both in photosystem I) had decreased accumulation specifically in group S in D at T1 and across treatments at T2, respectively (Appendix A).

We also performed GO enrichment analysis (at geneontology.org) to interpret functions of DEPs in common between: (i) groups defined by flag leaf size (58 DEPs), (ii) applied stresses (115 DEPs), and (iii) time points (152 DEPs) (Appendix A). DEPs shared across all genotypes with different flag leaf sizes were highly enriched (fold enrichment above 100) in terms associated with regulation of peptidase/endopeptidase, hydrolase, and proteolysis, mostly in a negative manner (biological process); this was generally confirmed within the molecular function categories (Appendix A). No significant results were observed for DEPs in common among D, H and HD (Appendix A), whereas numerous overrepresented GO terms were found for DEPs significant over time (Appendix A). Among DEPs shared between T1 and T2, the terms “positive regulation of cellular amide metabolic process” and “positive regulation of translation” were primarily enriched in the biological process category, while ribosome-related terms showed the greatest fold enrichment in cellular component; notably, approximately twofold more ribosomal DEPs were found in group L than in other genotypes (Appendix A). Additionally, we were interested in uncovering the functions of proteins whose levels were exclusively affected by the combination of drought and heat (62 DEPs), as well as those affected specifically at the early (104 DEPs) or late time point (157 DEPs). GO enrichment analysis indicated that, in the first case, terms associated with RNA splicing had the highest fold enrichment in the biological process category (Appendix A). Time-specific DEPs were enriched more numerously at T1 in terms related to spliceosome in both biological process and cellular component categories, including “Prp19 complex” (Appendix A).

### 3.2. Expression Co-Reaction Network

Co-reaction network analysis for DEPs identified in at least two contrasts revealed 11 modules (M1–M11) containing from 5 to 14 DEPs (Table 2 and Appendix A).

Overall, this analysis showed greater similarity of protein regulation between genotypes of groups S and M in response to stress treatments and some distinctiveness of group L behavior. However, considering the total DEPs identified in this study, the principal coordinate analysis (PCoA) of log_2_(FC) values for DEPs (significant in at least two comparisons) revealed that the proteomic reaction in group S was rather distant from the others (Appendix A). For instance, protein F2DEL5 (predicted protein) was not affected by applied stresses in genotypes of group S, whereas in group L it was upregulated in all contrasts (except for T1, HD) (Appendix A).

### 3.3. DEPs Classified According to Pfam Annotation

To more accurately characterize the functional domains/families of proteins induced by applied stresses, we analyzed the sets of DEPs with respect to Pfam annotation (Interpro database). Overall, in our proteomic dataset, we identified 656 Pfam entries, of which 256 corresponded to 173 DEPs (Appendix A); notably, one Pfam entry may be linked to several proteins of different behavior. Next, we extracted Pfam identifiers (Pfam IDs) that appeared in the annotation of a DEP or set of DEPs detected for: (i) all genotypes of different flag leaf sizes (S, M, L), (ii) applied stress treatments (D, H, HD), and (iii) two time points (T1, T2). Then, among them, we selected Pfam IDs whose definition may indicate that the related DEPs were involved in mediation of the response to stress (Appendix A). We found six Pfam IDs associated with the EF-hand domain; there were three DEPs containing this domain, and all were uncharacterized/predicted proteins. Interestingly, M0Z828 was upregulated in genotypes of groups S and M in D, HD at T2, and also in group L, but at the earlier time point. Curiously, protein F2DMV7 was exclusively downregulated in group L in HD at T2, whereas F2EJV2 showed a decreased level only in group M in HD at T1. In turn, the unknown protein F2DEL5 mentioned above had three Pfam IDs annotated to the DnaJ domain. Furthermore, we found three Pfam IDs annotated to the HSP domain or HSP family, namely, HSP20, HSP70, and HSP90, which were present in six DEPs (all described as predicted proteins, with one exception). One of them, F2E4C2, showed increased accumulation in HD at T2 across all genotypes of different flag leaf sizes, whereas F2D884 reacted more specifically; namely, it was upregulated by H in group S at T2 and by D in group L at T1, but downregulated by D in group M at T2. Otherwise, F2D2B6 showed reduced accumulation only in D, relative to the control, in genotypes of group S at T2 and in group L at both time points. In turn, protein F2D5G5 was exclusively upregulated in genotypes of small flag leaf size at T1 across stress conditions. Next, we detected the presence of the C/N-terminal domain of glutathione S-transferase (six Pfam IDs) in the annotation of three DEPs: two of them were known as glutathione transferases and one was an uncharacterized protein. They changed the regulation status primarily in genotypes of group L (and also in S in H at T2). That is, M0Y1I9 had increased accumulation across treatments at T2, which was similar for Q8VWW3, but excluding HD. In contrast, Q3MUP2 was exceptionally downregulated in response to HD in group L at T2. Seven Pfam IDs representing tetratricopeptide repeat were found in the annotation of SGT1 protein (Q8W516). Remarkably, this protein showed a switch of regulation status over time; it was downregulated and upregulated in group S in D at T1 and T2 (also in HD), respectively. Analysis of Pfam entries indicated that five DEPs of unknown function (excluding 30S ribosomal protein S2–A0A218LNH5) were members of the ribosomal protein family (five Pfam IDs).

### 3.4. Relationship between Gene and Protein Regulation

Among all identified proteins, 1073 had an assigned gene identifier; out of those, 306 were differentially expressed in the experiment (Appendix A). Based on transcriptomic data (reported previously), we found 50 cases where the gene corresponding to the DEP was also differentially expressed (DEG) in at least one (the same or not) comparison. The majority of genes (32 DEGs) were downregulated primarily in genotypes of group S at T2 in D and HD, while at the protein level, the proportions of up- and downregulation events were rather equal (Appendix A). Intriguingly, we found 14 gene–protein pairs (presumably the gene and its encoded protein) whose expression was significantly modified in response to applied stresses in at least one and the same contrast. Overall, within these pairs, the direction of gene and protein regulation was consistent (with “upregulation” predominating), especially in response to D and HD treatments, mostly at T2 (Table 3 and Appendix A).

The largest number of common DE events was observed for pairs HORVU1Hr1G070690-A0A287G4N5 (gene annotated to PEBP-like superfamily according to Interpro database), HORVU1Hr1G065150-Q40036, and HORVU7Hr1G113200-M0Z828 (gene annotated to EF-hand domain containing protein according to Interpro database), primarily in genotypes of groups S and M. There were two dehydrins, and the upregulation of A0A287S8C1 was confirmed at the transcriptomic level only in genotypes of group M in D at T2, whereas A3RHE3 (described above) was overexpressed together with the corresponding gene in genotypes of group M in D and HD at T2, but in group S at T2, they showed the opposite behavior in response to HD (Table 3). The gene HORVU4Hr1G054870 was annotated to “response to stress” and, along with the paired protein, was upregulated in genotypes of groups S and M in D at T2. Two chlorophyll a/b binding proteins and the genes encoding them were downregulated in genotypes with a medium flag leaf size in D at T2; notably, the DE events were substantially more numerous at the transcriptomic level than at the proteomic level. In contrast, for BURP domain-containing protein, almost four times as many DE events (significant especially at T1) were found than for the transcript of the linked gene, and they were upregulated jointly only in genotypes of group S in D at T2. Among gene–protein pairs regulated simultaneously in the same contrast, two genes encoding enzymes had *Arabidopsis* orthologs. The gene HORVU1Hr1G091600 (ortholog of *APL2*, *APL3*, and *APL4* genes) with linked protein O04896 and the gene HORVU7Hr1G047700 (ortholog of *FDH1* gene) with linked protein F2CWR0 were upregulated in groups L and M, respectively, both pairs in response to HD at T2. Meanwhile, a third gene (HORVU5Hr1G001180) encoding an enzyme (i.e., a lipoxygenase) was downregulated along with protein (M0Y1R9) in flag leaves of genotypes of group S in D and HD at T2 (Appendix A).

Finally, we constructed a network of proteins and genes substantially correlated with respect to their expression change, whose regulation was affected by stress stimuli (Figure 3).

In total, there were 10 DEPs significantly associated with 40 DEGs (Appendix A). The largest number of significant correlations (12) was found for the protease inhibitor Q40036. The gene HORVU3Hr1G067350 (annotated to B3 domain-containing transcription factor FUS3 according to Ensembl Plants) exhibited the strongest correlation with this protein. Ten genes were associated with F2D3S4, described as low-temperature-induced 78 kDa/65 kDa protein (Interpro database). This protein showed the most significant correlation to the gene HORVU4Hr1G050930 (cytochrome P450), and also to three genes annotated to the SANT/Myb domain (HORVU0Hr1G007770, HORVU0Hr1G010940, and HORVU7Hr1G083690 being orthologs of *Arabidopsis GLK2* genes). Interestingly, we identified the relationship between dehydrin (A0A0S1LGH4) and expression change of four genes: two were annotated to ankyrin repeat, one was related to ATC domain, and one gene encoded LEA2, which was also associated with the regulation of the F2D3S4 protein. We found that F2E325–glutaredoxin protein (according to Interpro database) was correlated with expression of the gene HORVU1Hr1G021650 engaged in defense response through thaumatin.

## 4. Discussion

Over the years, substantial efforts have been made to decipher the molecular response of canopy elements and the root system of barley to stress stimuli; however, little attention has been paid to characterizing the flag leaf, especially in large-scale studies. In this study, the proteomic landscape was investigated in the same experimental model as used in previous transcriptomic profiling of barley flag leaf (see Mikołajczak et al. [22]).

Four types of plant responses to combined stress were proposed by Prasch and Sonnewald [29]: (i) additivity or (ii) synergy of individual stress responses, (iii) dominance of response to one stressor over another, or (iv) reactions that completely differ from the response to a single stress. Overall, the obtained results indicated that the proteome remodeling induced by combined stress was more similar to that observed under the single stress of drought than under heat stress alone. Thus, the response to HD did not involve the synergy or additivity of the effects of individual stresses. Evidently, drought was more severe for plants than heat in our study. However, it also cannot be unequivocally asserted that plants preferentially exhibited the dominance strategy during the co-occurrence of stress, since despite a large overlap between D and HD, substantial proportions of stress-specific DEPs underwent unique reactions to each stress. Indeed, among the above-mentioned strategies of plant response to multiple stressors, idiosyncratic strategy in various plant species have most often been reported by researchers [29]. 

We proved that the proteomic signature was time-dependent and that prolonged exposure to stress induced more changes in protein abundance than shown early in the stress period. However, more than half of the DEPs overlapped between the time points and the core of the early and late responses to stress was concentrated on particular functions since numerous enriched GO terms were found for shared DEPs, primarily associated with translation. This was similar for proteins whose expression was induced exclusively in T1. Immediate remodeling of translational machinery is fundamental for plants to adjust effectively to constrained conditions [4]. In our study, proteins involved in translational processes were rapidly affected, which continued throughout stress application. Apparently, this reaction was insufficient for the plants’ adjustment to unfavorable conditions in the following days of stress; therefore, genome-wide proteomic changes were intensified in T2 specifically.

Proteome-wide differential analysis revealed the dominance of upregulation events under stress, regardless of the treatment type and flag leaf morphology. This general finding is not surprising since the massive accumulation of proteins of various functions has been well documented under stress stimuli [30,31]. Interestingly, we found that stress-induced proteins were the most numerous in barley plants with a large flag leaf size. This may suggest that the larger flag leaf blade, the greater perception of stress and, consequently, the more numerous changes in protein abundance. However, this was not observed at the transcriptomic level in our previous study, where gene expression was affected the most in genotypes with a small flag leaf [22]. It is possible that translational and post-translational processes strongly differentiate analyzed genotypes with different flag leaf blades, especially during stress perception, which could be one of the explanations for the mentioned discrepancy. Our assumption becomes more reliable by noting that ribosomal-related proteins, as the basic translational machinery [32], were more numerously affected by stress in plants with a large flag leaf (ca. 2-fold) than in others. Second, the GO enrichment analysis revealed that terms associated with spliceosome were strongly overrepresented within early responsive DEPs and within those specific to combined stress. Interestingly, only 20% and 8% of them were shared between genotypes of groups L and S, respectively. This may indicate that there were substantial discrepancies in RNA splicing between the genotypes with different flag leaf sizes. Presumably, in one group of genotypes, alternative splicing may occur to a greater extent under stress conditions, while in others, constitutive splicing predominantly occurs. This hypothesis warrants further investigation.

In a global sense, the proteomic reactions of genotypes L and S to applied stresses were distant from each other, which was confirmed by PCoA and expression co-reaction network analysis. In-depth interpretation of the results also showed numerous cases of exclusive protein behavior under stress stimuli in genotypes with different flag leaf blades, for example, proteins involved in the photosynthetic apparatus. Previously, we identified the relationship between flag leaf size and photosynthetic parameters Fv_Fm along with Pi_ABS and found that photosynthesis-related genes were also differentially expressed under stress between genotypes with different flag leaf [22]. Herein, we found some differences in the regulation of components of photosystems PSI and PSII (absorbing light in different wavelength ranges; [33]) between genotypes of groups S and L. Disturbances in the large flag leaf were related to the first system of electron flow (PSII) under HD. In turn, group S was perturbed in PSI under drought, which may cause limitations in NADPH^+^ synthesis and disruption of the Calvin cycle. Meanwhile, ferredoxin (Fd), the acceptor of electrons from PSI, was simultaneously upregulated in group S. Increased accumulation of Fd may be due to its role in mediating redox signaling [34]. It can be speculated that the lack of a negative effect of drought on proteins of PSII in group S may have resulted from overexpression of the HORVU2Hr1G090070 gene encoding a constituent of the OEC complex (putatively PsbQ), an essential part of photosystem PSII, as demonstrated in our other study [22]. Some evidence has been unearthed revealing that an increased content of OEC-related proteins may confer better drought tolerance in barley plants [35,36]. In turn, it can be claimed that plants with a large flag leaf differed from others in their response to oxidative stress in a manner mediated by glutathione transferase. The upregulation of M0Y1I9 and Q8VWW3 may indicate the effective detoxification in group L since glutathione transferases are well known to be involved in coping with increased levels of free radicals [37]. On the other hand, these enzymes also catalyze a wide range of metabolic reactions not related to detoxification, so their decreased accumulation may also occur under unfavorable conditions [38]. Notably, SGT1 protein exhibited specific behavior in genotypes of group S; its abundance was decreased at an early stage in drought but increased with prolonged water scarcity. This conserved eukaryotic protein binds to the molecular chaperone HSP90 [39], which could indicate that its overaccumulation is a rapid response to stress. On the other hand, Gray et al. [40] suggested that SGT1 was required for SCF (Skp1-cullin 1-F-box) complex functioning in *Arabidopsis*. Since this complex participates in ubiquitination [41], SGT1 may act as one of the regulators of the degradation of a multitude of proteins. Indeed, SGT1 may have the ability to interact with other proteins because it contains a tetratricopeptide repeat motif, as we showed using Pfam annotation; it mediates protein–protein interactions and thus contributes to the formation of multiprotein complexes [42].

Apart from the stress-responsive proteins whose expression was specifically induced, there were also DEPs shared between genotypes of different flag leaf sizes, stress variants, or time points. The DEPs identified across the applied stresses may underlie the universal stress response and thus constitute promising biomarkers for improving cereals’ resistance to environmental hazards [43]. One of the strongest candidate groups in our study for universal stress-responsive proteins was dehydrins. LEA proteins, including dehydrins, play protective roles through stabilizing enzymes and membranes under multiple abiotic stresses [10]. Hence, Q5D5Z6 and F2DH00, identified in our study, appear to be robust candidates for biomarkers of resistance to multiple stressors. Likewise, the early responsive aquaporin O48518, which facilitates the transport of water and small uncharged molecules across biological membranes [44], may be a useful candidate conferring osmotic stress tolerance to barley, as suggested by Alavilli et al. [45] at the gene expression level.

To further understand the biological functions of unknown stress-induced proteins, we used Pfam annotation. This enabled the identification of several uncharacterized proteins that can mediate stress responses. For instance, none of the known barley calmodulins was differentially accumulated in our study, but three DEPs (M0Z828, F2DMV7, F2EJV2) containing an EF-hand domain were found. This domain is present in signaling proteins and binds calcium [46]; thus, we proposed that these three DEPs are Ca^2+^ messengers in barley, which presumably represent calmodulins. Interesting behavior of these proteins was observed in plants with a large flag leaf. Specifically, M0Z828 was overaccumulated in group L earlier than in other genotypes in response to D and HD; therefore, we believe that the calcium-mediated transmission of stress signal was activated more rapidly in L plants. In contrast, F2DMV7 was downregulated in late HD in group L. Numerous reports have highlighted the importance of calmodulins during the adjustment of plants to adverse conditions (reviewed by Bergey et al. [47]); thus, their upregulation under stress was expected in our study. However, it has also been suggested that, during intense and prolonged stress, plants start to orientate primarily toward surviving, not developing [48]. One possible mechanism regulating such reorganization is the change in calcium signal dynamics leading to a reduction in cellular activity because the transition of Ca^2+^ affects multiple metabolic pathways, influencing cell growth and proliferation [49]. Supposedly, barley plants with a large flag leaf may implement this strategy by downregulating F2DMV7 under the late co-occurrence of drought and heat. Next, based on Pfam annotation, we defined five uncharacterized DEPs as functioning as HSPs. They operated in different ways in our experiment, being up- or downregulated in response to the various stresses. Some of them responded to early stress and others to late stress, while F2D2B6 was affected by drought in group L independent of the stress duration. In turn, F2D5G5 showed a stress-universal response in genotypes of group S. Notably, we found that the uncharacterized protein F2DEL5 contained a DnaJ domain, suggesting that it also acts as a chaperone, primarily in group L. Such ambiguous behavior of numerous HSPs was also observed at the transcriptomic level (even more numerous) in our corresponding study [22]. This protein family has been implicated not only in responses to temperature stimuli, as thought initially, but also to other environmental hazards [50], and in overall plant growth and development as well [51]. These facts can explain the erratic activity of HSPs in our study.

Next, we integrated the global proteomic and transcriptomic data extracted from flag leaves of barley. To our knowledge, no previous studies have attempted to analyze quantitative proteomic datasets from barley flag leaf under combined heat and drought and integrate them with mRNA abundance. There were discrepancies in the regulation of expression of most genes and assigned proteins, indicating that mRNA expression cannot serve as an accurate predictor of protein abundance at the time of measurement. The protein content is multidimensionally determined; it depends on the transcription efficiency and mRNA stability but also on the regulation of translation and protein degradation [52]. Understanding the transcriptome clearly facilitates the exploration of plant genomes. Meanwhile, proteins contribute more directly to managing cellular functions than transcripts [53]. We identified several clear examples of coincident expression of a gene at the transcriptomic and proteomic levels, mainly under D and/or HD. Interestingly, some of them appear to be involved in stress-adaptive pathways. For instance, we identified stress-induced upregulation (protein along with encoding gene) of two dehydrins as well as of A0A287P420 and of A0A287LIR7. A0A287P420 contains a Bet_v_1 domain, which belongs to the pathogenesis-related PR10 protein family. Apart from the known role of PR10 in defense against pathogens, its enhanced expression under various abiotic hazards in different plant species, resulting in improved stress tolerance, has also been documented [54]. Similarly, it has been proposed that formate dehydrogenase, encoded by the *FDH1* gene in *Arabidopsis*, confers tolerance to biotic and abiotic stresses [55,56]. Curiously, we found that its barley ortholog, HORVU7Hr1G047700, was upregulated in flag leaf in response to HD, along with the protein that it encodes. Thus, we consider both mentioned gene–protein pairs to be promising tools for the engineering of barley to achieve biotic and abiotic stress resistance. In turn, the protein A0A287LIR7 comprises a BURP domain and is encoded by a gene in barley, HORVU3Hr1G069650, that is the ortholog of the *RD22* gene of *Arabidopsis*—a known reference gene for drought stress response. Moreover, BURP domain has been suggested to participate in signaling pathways of abscisic acid (ABA) and salicylic acid (SA), which are key stress-responsive phytohormones [57,58]. It should be pointed out that only a few BURP domain-related studies have been performed in barley. Next, we found possible modulators of sugar-mediated plant response to stress, namely, HORVU1Hr1G091600 and O04896. This gene is the ortholog of *Arabidopsis APL* genes, and it was upregulated along with O04896 in our experiment. *APL* genes are involved in starch biosynthesis, especially in photosynthetic tissues [59]; therefore, the protein O04896 supposedly plays a similar role in barley flag leaf. Starch is the reservoir of carbohydrates, and its remobilization induced by threat supplies the energy and carbon, allowing the plant to mitigate the effects of stress [60]. Since for the above-mentioned examples the convergent reaction of transcript and protein was captured at the same time of measurement, we assumed that the plant required their continuous, not occasional, expression to produce the functional proteins during stress. We speculated that this response was more basic and guaranteed a rapid, direct reaction to stress, not one entangled in complicated regulatory cascades. Thus, the selected gene–protein pairs may constitute the most promising candidates for improving the tolerance of barley to abiotic stresses in the future.

To further explore the regulatory network through which barley adapts to drought and heat, we examined the correlations between changes in the expression of genes and proteins induced by the applied stresses. We did this by analyzing a subset of the correlation network constructed for expression of 1037 genes and 413 proteins restricted to gene–protein pairs characterized by a substantial correlation, or adjacency in terms of the WGCNA method. It should be noted that WGCNA is aimed primarily at the identification of correlated groups (modules) of traits. As in any correlation analysis, the interpretation of particular relationships between pairs of traits (network edges) may be affected by the presence of spurious correlations arising due to the influence of other variables. This problem could be dealt with by the application of partial or conditional correlations. However, the reliable conditioning of correlations, in this highly dimensional situation, would probably require more observations available for computations than eighteen contrast estimates used in our case. Interpretation of the correlation network revealed several potential candidates affecting the upregulation of serine-type proteinase inhibitor Q40036, most likely the Bowman–Birk inhibitor (BBI) family. The strongest correlation between genes and proteins was observed for the gene encoding the FUS3 transcription factor. Indeed, this TF has been reported to be directly targeted to many other TFs or indirectly via relationships with genes encoding microRNAs [61]. BBI was initially considered to protect against pathogenic infection by inhibiting exogenous proteases, but recently, it has also been suggested to mitigate oxidative stress damage [62]. Protease inhibitors may affect numerous proteins and, notably, Q40036 showed increased accumulation in most “contrasts” defined in our experiment. Hence, it can be claimed that it constitutes a good biomarker of the universal response to stress. Next, we identified some connections between stress-responsive components of the photosynthetic pathway. The upregulation of the protein F2D3S4 involved in ABA-mediated stress signaling was found to be related to the expression of genes annotated to cytochrome P450 and SANT/Myb domain (ortholog of the *GLK2* gene of *Arabidopsis*). These relationships appear to be justified in stressed flag leaf of barley because the mentioned genes are associated with photosynthesis [63,64], whereas an increased content of ABA in response to stress stimulates stomatal closure and reduces the efficiency of photosynthesis [65]. The network of correlations showed that the barley dehydrin may interact with other proteins through genes encoding protein-containing ankyrin repeats. This motif is well known to mediate protein–protein interactions and cell–cell signaling [66], and it can activate the transcription of various stress-related genes, conferring better plant tolerance to the threat [67]. We also detected a correlation between glutaredoxin and a gene annotated to thaumatin. Glutaredoxin is inducible by SA under adverse conditions and catalyzes the reduction in disulfide transitions [68]; notably, the interplay of SA with redox signals has been documented [69]. Thaumatin belongs to the pathogenesis-related protein family PR5, and it has been demonstrated that the accumulation of members of this family increases during the SA-induced defense response [54,70]. In this context, it can be postulated that we identified the regulatory components of the oxidative stress response mediated by SA.

## 5. Conclusions

A large-scale proteomic study enabled the identification of numerous flag leaf size-specific alternations in protein accumulation under drought, heat or both, including proteins involved in the photosynthetic apparatus. Universally responding proteins to stress were also found, including dehydrins. Using Pfam annotation, we proposed the putative function of several unknown proteins that can mediate responses to stress stimuli, such as calmodulins and HSPs. Finally, the confrontation of gene and protein expression highlighted the important elements of various regulatory pathways of the oxidative stress response, including abscisic acid- and salicylic acid-mediated reactions. The identified stress-induced proteins may serve as useful biomarkers of barley resistance to multiple abiotic stresses and could be exploited for future breeding of cereals.

## Figures and Tables

**Figure 1 cells-12-01685-f001:**
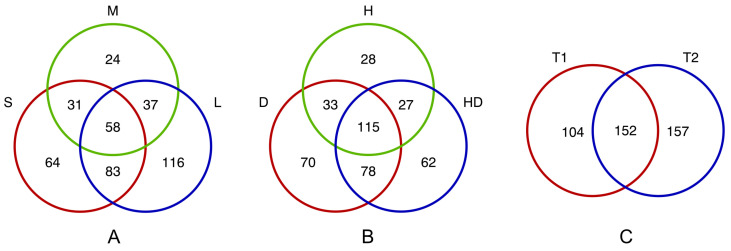
Venn diagrams visualizing the number of differentially expressed proteins in flag leaves, specific and shared between: (**A**) barley genotypes assigned to groups of different size of flag leaf (S, M, L), (**B**) applied stress treatments (D, H, HD), and (**C**) time points (T1, T2).

**Figure 2 cells-12-01685-f002:**
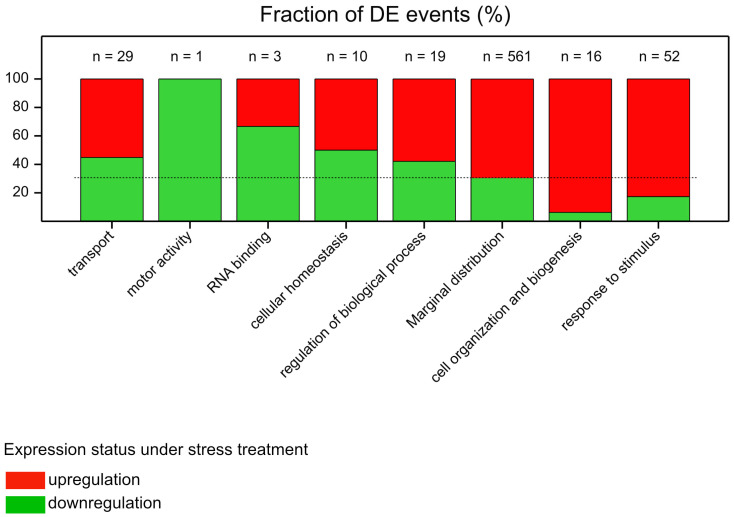
GO terms for which the distribution of assigned down- and upregulation DE events was mostly different than the marginal distribution.

**Figure 3 cells-12-01685-f003:**
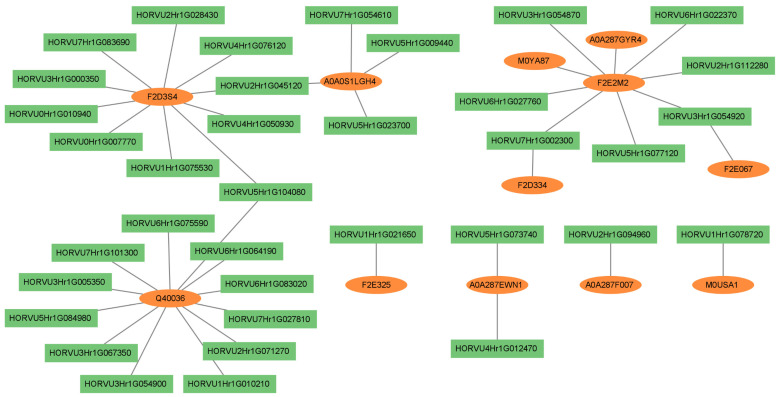
The network of differentially expressed genes and proteins with substantially correlated expression profiles (gene–protein pairs with correlation coefficient > 0.6, *p* < 0.01).

**Table 1 cells-12-01685-t001:** Number of down- and upregulated proteins in genotypes classified to different groups according to flag leaf size, at time points T1 and T2, in comparisons between drought (D), heat (H), and their combination (HD) vs. control.

Group According to Flag Leaf Size	Regulation Status	T1			T2		
		D	H	HD	D	H	HD
Small (S)	Down	13	7	11	27	9	30
	Up	28	38	60	72	28	63
	Total	41	45	71	99	37	93
Medium (M)	Down	5	3	12	44	3	13
	Up	20	6	16	42	15	54
	Total	25	9	28	86	18	67
Large (L)	Down	34	24	16	19	32	54
	Up	106	50	39	46	50	61
	Total	140	74	55	65	82	115

**Table 2 cells-12-01685-t002:** Co-reaction modules containing DEPs significant in at least two contrasts.

Group According to Flag Leaf Size	Time Point	Treatment	M1	M2	M3	M4	M5	M6	M7	M8	M9	M10	M11
S	T1	D	−0.49	−0.83	−0.50	0.40	−0.46	0.05	−0.15	−0.21	−0.27	−0.28	−0.20
		H	−0.52	−0.08	−0.16	0.65	−0.93	0.29	−0.34	−0.15	−0.22	0.40	−0.46
		HD	−0.81	0.96	0.68	0.02	−0.77	0.18	0.20	−0.38	0.26	1.45	−0.91
	T2	D	1.57	−0.31	−0.39	−1.12	1.44	−0.42	0.18	0.08	−0.38	−0.91	1.34
		H	0.10	−0.57	−1.17	0.21	0.66	−0.18	0.38	0.27	−0.18	−0.43	0.89
		HD	1.53	−0.78	−0.94	−1.04	1.54	−0.47	−0.30	−1.31	0.05	−0.84	1.49
M	T1	D	−0.68	0.28	−0.28	0.25	−0.33	0.10	−0.06	0.29	0.06	0.05	0.23
		H	−0.74	0.31	−0.21	0.03	−0.19	−0.15	0.08	−0.19	0.10	0.16	0.13
		HD	−0.37	0.11	−0.28	0.04	−0.49	0.10	−0.16	0.17	−0.07	0.06	0.16
	T2	D	0.52	−2.01	−1.48	−0.31	1.49	0.22	−0.43	−0.05	−0.10	−1.22	0.81
		H	−0.42	−0.70	−0.79	0.28	0.39	−0.11	−0.29	0.70	0.14	−0.28	0.19
		HD	1.70	−0.93	−0.53	−0.19	1.41	−0.03	−0.08	0.25	0.21	−0.79	1.24
L	T1	D	−0.54	1.94	2.01	0.21	−0.45	−0.07	0.92	−0.55	0.30	1.53	−0.59
		H	−0.74	1.06	1.31	0.42	−1.05	0.05	0.16	−0.81	0.35	0.94	−2.12
		HD	−1.01	0.38	0.47	−0.03	−0.91	0.47	0.06	−0.44	0.18	0.29	−0.44
	T2	D	−0.34	0.78	0.92	0.00	−0.64	−0.36	−0.05	1.00	−0.07	0.44	−0.50
		H	−0.22	0.99	0.99	−0.14	−0.37	0.02	−0.09	0.88	−0.32	0.59	−0.89
		HD	1.45	−0.58	0.35	0.32	−0.36	0.32	−0.04	0.45	−0.03	−1.15	−0.37

**Table 3 cells-12-01685-t003:** Gene–protein pairs with significant alternation of regulation in a coincident comparison out of 18 defined contrasts.

Gene Identifier	Protein Identifier (Protein Name)	Number of Common DE Events	Regulation Status (Up/Down)		
			Gene	Protein	
HORVU1Hr1G065150	Q40036 (protease inhibitor)	4	up	up	
HORVU1Hr1G070690	A0A287G4N5 (uncharacterized)	6	up	up	
HORVU1Hr1G091600	O04896 (glucose-1-phosphate adenylyltransferase)	1	up	up	
HORVU2Hr1G079180	F2DH67 (predicted protein)	2	down	down	
HORVU3Hr1G069650	A0A287LIR7 (BURP domain-containing protein)	1	up	up	
HORVU4Hr1G054870	A0A287P420 (Bet_v_1 domain-containing protein)	2	up	up	
HORVU4Hr1G087760	A0A287Q1Q5 (AAI domain-containing protein)	1	down	up	
HORVU5Hr1G001180	M0Y1R9 (lipoxygenase)	2	down	down	
HORVU5Hr1G092100	A0A287S8C1 (dehydrin)	1	up	up	
HORVU5Hr1G109260	A0A287SNX7 (chlorophyll a-b binding protein)	1	down	down	
HORVU6Hr1G016890	A0A287TFZ0 (chlorophyll a-b binding protein)	1	down	down	
HORVU6Hr1G084070	A3RHE3 (dehydrin 4)	3	up	down	up
HORVU7Hr1G047700	F2CWR0 (formate dehydrogenase)	1	up	up	
HORVU7Hr1G113200	M0Z828 (uncharacterized)	4	up	up	

## Data Availability

The data underlying this article are available in the article and in Appendix A.

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
