# Peer review of "Global Proteome Profiling Revealed the Adaptive Reprogramming of Barley Flag Leaf to Drought and Elevated Temperature"

_cells, 2023, doi:10.3390/cells12131685_

Round 1
Reviewer 1 Report
In this paper, a large number of stress releated proteins were obtained based on global proteome profiling. However, there are still the following problems that need to be revised
1. How to define the leaf size? small–S, medium–M, 110 large–L? It is recommended to listed the photos of different sizes of leaf into an online document.
2. The paper lacks necessary qRT-PCR experiments to verify the accuracy of the data.
3. Table 3, It is recommended to list homologous protein names rather than assigned protein ID。
Reviewer 2 Report
The manuscript reports an interesting and well-conducted study demonstrating global alterations of the protein and gene expressions associated with the exposition of plants to different stress factors.
The statistical analysis presented seems to be sensible, however, I remark that the gene and protein expression network presented in Figure 3 is constructed using statistically significant correlations. My concern in this part of the analysis is that significant correlations might be generated by spurious correlations in the sense that some reported correlations between two variables might occur because these two variables are both highly correlated with a third variable so that conditionally on the third variable, the two variables would be non-correlated. This issue is especially problematic when the third variable (the one that renders the two putatively correlated variables) is also present in the network. A way to circumvent this problem would be to construct the expression network using partial correlations instead of correlations, as it is usually done when using graphical models.
I suggest that the authors re-construct the expression network presented in Figure 3 using partial (or conditional) correlations or make a warning remark regarding the possibility that some of the reported correlations might be spurious. Technically, the partial covariances can be obtained by inverting the covariance matrix.
Reviewer 3 Report
The MS with the title, “Global proteome profiling revealed the adaptive reprogramming of barley flag leaf to drought and elevated temperature” by MikoÅ‚ajczak et al. is a fine effort to elucidate the protein changes in one of the most important organ (flag leaf) of barley in response to drought, heat and its combination. The authors used state of the art liquid chromatography and high-resolution mass spectrometry to identify stress-responsive molecular mechanisms in barley adaptation to multiple abiotic stresses. The authors provided a detailed analysis of the role of the putative functions of the otherwise unknown proteins (calmodulin like proteins). The authors also constructed a correlation network between transcripts and proteins performance to provide a framework of stress adaptive pathways in the flag leaf.
There are certain drawbacks in the MS as well. One is its length. It is too lengthy. It takes the readers a lot of time to read and extract important information from the MS in question. Likewise, try to conclude your MS in fewer lines than it has been presented at the moment.
I have made some comments in the attached file for you to address.

Round 2
Reviewer 1 Report
I have no other questions